# Newly Diagnosed Right Coronary Artery Aneurysm in an Adult with Recent Coronavirus Disease 2019 Infection

**DOI:** 10.3390/diagnostics12040815

**Published:** 2022-03-26

**Authors:** Jeong-Hun Shin, Sun Kyun Ro

**Affiliations:** 1Division of Cardiology, Department of Internal Medicine, Hanyang University Guri Hospital, Hanyang University College of Medicine, Guri 11923, Korea; cardio.hyapex@gmail.com; 2Department of Thoracic and Cardiovascular Surgery, Hanyang University Guri Hospital, Hanyang University College of Medicine, Guri 11923, Korea

**Keywords:** coronary artery aneurysm, coronavirus disease 2019, adult

## Abstract

It is well known that coronavirus disease 2019 (COVID-19) can affect the vascular endothelium; however, coronary artery aneurysm complicated by COVID-19 in adults has not yet been reported. Herein, we report a case of a newly developed right coronary artery aneurysm in an adult with recent COVID-19 infection. A 66-year-old man was referred for surgical intervention of the right coronary artery aneurysm. His previous coronary angiograms performed 17 months prior revealed no evidence of coronary aneurysm. However, he was confirmed as having COVID-19 four months prior and thereafter newly diagnosed with right coronary artery aneurysm. The further evaluation of the impact of COVID-19 on the occurrence of coronary artery aneurysm may be necessary, considering the alleged correlation between COVID-19 and vascular complications.

## 1. Introduction

As coronavirus disease 2019 (COVID-19) cases have been accumulated, various associated complications have been reported. It is well known that COVID-19 can affect the vascular endothelium through the previously published studies. Multicenter case series concerning multiple system inflammatory syndrome involving coronary arteries in children and adolescents have also been published [1,2]. However, coronary artery aneurysm complicated by COVID-19 in adults has not yet been reported. Herein, we report a case of a newly developed right coronary artery aneurysm in an adult with recent COVID-19.

## 2. Case Presentation

A 66-year-old man on hemodialysis was admitted to our hospital for further evaluation of newly developed chest pain. He had diabetes mellitus, hypertension, and a history of cerebral infarction. He had a history of coronary stent implantation in the left anterior descending artery at its mid- and distal portions and in the left circumflex artery (LCx) at its proximal portion 17 months prior. On coronary angiograms performed at that time, there were no findings suggestive of coronary artery dilatation, aneurysm, or ectasia (Appendix A). He was regularly followed up with dual antiplatelet therapy thereafter. Four months prior to this presentation, a polymerase chain reaction test confirmed the diagnosis of COVID-19, for which he was admitted to another hospital because of combined pneumonia and his comorbidities for about a month. One month prior, he had received his first COVID-19 vaccination (Vaxzevria; AstraZeneca, Oxford, UK).

To investigate the association between newly developed chest pain and coronary lesions, the patient underwent diagnostic coronary angiography, which revealed a large coronary artery aneurysm with a diameter of 8.6 mm at the proximal portion of the right coronary artery (RCA) and multiple stenotic lesions on the RCA and LCx (Appendix A). Transthoracic echocardiography showed an ejection fraction of 40%, with hypokinesia of the base to mid inferior wall and basal inferolateral wall, and akinesia of the mid inferolateral wall. Magnetic resolution cerebral angiography showed no aneurysmal changes to the cerebral arteries. The initial C-reactive protein level was within normal limits. The patient and the heart team decided to perform surgical interventions to treat him for both multiple stenotic lesions and the coronary aneurysm simultaneously. Coronary artery bypass grafting was performed using an on-pump beating strategy without cardiac arrest. One saphenous vein graft was anastomosed to the distal RCA, while the other was anastomosed to the obtuse marginal branch of the LCx. Suture ligation of the RCA aneurysm at its proximal and distal portions was also performed (Figure 1). Postoperative coronary computed tomographic angiograms obtained 7 days after surgery showed patent vein grafts and no contrast enhancement in the RCA aneurysm (Figure 2). His hospital course was uneventful. He was regularly followed up for 5 months after discharge without complications.

## 3. Discussion

Coronary artery aneurysm is a rare disease entity known to be caused by atherosclerosis, Kawasaki disease, congenital anomaly, mycotic disease, connective tissue disease, and arteritis [3]. As clinical data concerning COVID-19 caused by severe acute respiratory syndrome coronavirus 2 continue to accumulate, it is well known that COVID-19 can primarily involve the respiratory system and vascular structures, especially the vascular endothelium [4]. Indeed, several multicenter case series of multisystem inflammatory syndrome involving the coronary artery in children and adolescents have been published [1,2]. To our knowledge, however, coronary artery aneurysms associated with COVID-19 infection in adult patients have not been reported. In the current case, an RCA aneurysm that was not previously noted on coronary angiograms 17 months prior developed after a recent COVID-19 infection. Understandably, not every previous event is a cause of following outcomes. Moreover, hemodialysis patients are vulnerable to coronary atherosclerotic changes, the most common cause of coronary aneurysms in adults. Nevertheless, COVID-19 can affect the vascular endothelium, and the pathogenesis of coronary artery aneurysms is not yet well understood. Moreover, the RCA of the patient in the current case had been quite normal and not been touched 17 months prior. Therefore, the relationship between recent COVID-19 infection and newly developed coronary aneurysms could not be excluded. Clinical manifestations complicated by COVID-19 in adult patients might be somewhat different from those with multisystem inflammatory syndrome. In this regard, it would be important to collect more cases concerning this issue further.

Since most recent recommendations are based on limited case studies, treatment options for coronary artery aneurysms vary according to the anatomy of the affected coronary artery, combined anomalies, and the presence of coronary stenosis. Percutaneous interventions for coronary aneurysms have been reported in published case studies [5]. However, as in the current study, surgical interventions should be considered if concomitant coronary artery bypass grafting is necessary. Surgical techniques include ligation, resection, or open repair of the aneurysm [6,7]. Although the current case was not applicable, anti-inflammatory treatments such as intravenous immunoglobulins and steroids should be also considered if there were clinical evidence suggesting multisystem inflammatory syndrome [8].

## 4. Conclusions

Whether coronary artery aneurysms are caused by COVID-19 in adults is unclear. Nevertheless, the further evaluation of the impact of COVID-19 on the occurrence of coronary artery aneurysms may be necessary considering the alleged correlation between COVID-19 and vascular complications.

## Figures and Tables

**Figure 1 diagnostics-12-00815-f001:**
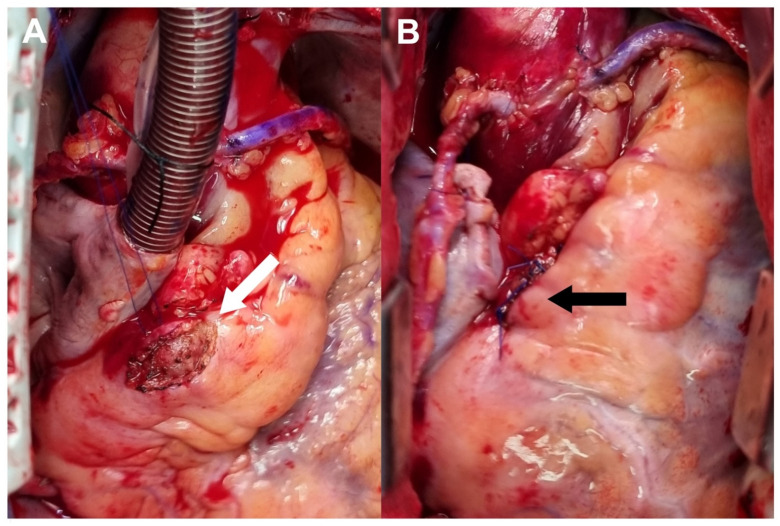
Operative Findings. (**A**) The right coronary artery was exposed at its proximal portion (white arrow). (**B**) Suture ligation was performed at the proximal and distal portions of the right coronary artery aneurysm (black arrow).

**Figure 2 diagnostics-12-00815-f002:**
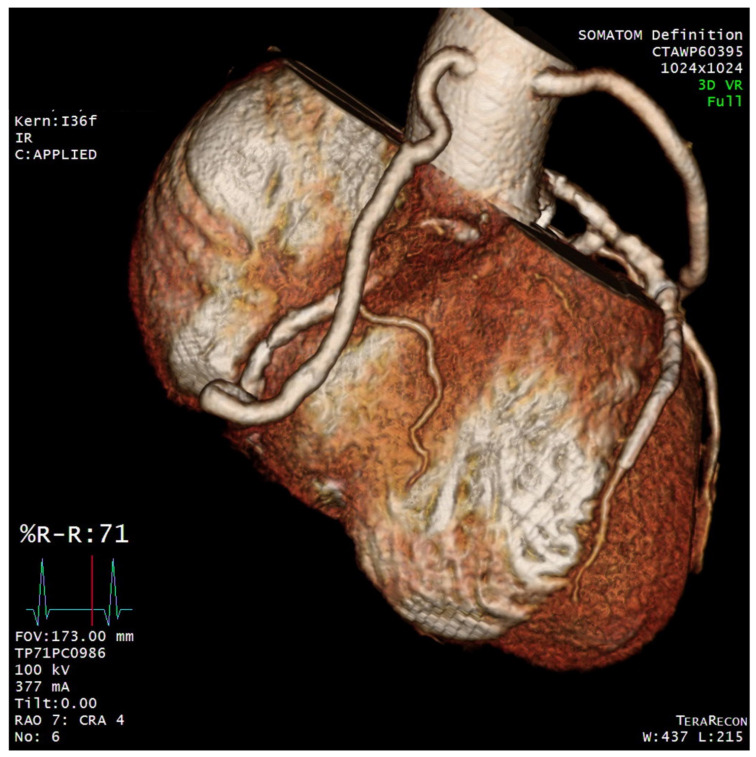
Postoperative Computed Tomography Finding. Computed tomography revealed that the bypass grafts were patent without contrast fill in the aneurysmal portion of the right coronary artery.

## Data Availability

Not applicable.

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
