# Peer review of "Newly Diagnosed Right Coronary Artery Aneurysm in an Adult with Recent Coronavirus Disease 2019 Infection"

_diagnostics, 2022, doi:10.3390/diagnostics12040815_

Round 1
Reviewer 1 Report
Thank you for giving me the opportunity to review this case report.
Please find my comments below:
- Page 1, line 25: There are no case reports of children but large cohorts for multiple centers and should be referenced.
- Page 3, line 66; Reference
- The authors stated that the aneurysm can be a consequence of the underlying disease as well as Covid, however difficult to tease out. There are reports of adults suffering from MIS –C syndrome however with no significant complications. This case could belong into that category.
- Was the patient investigated for MIS-C? Did he receive any anti-inflammatory treatment?
General comments: It is an interesting case, however it is difficult to prove whether the aneurysm was a consequence of the underlying condition or post Covid infection. That should be stated more clearly in the text.
-
- My recommendation is for publication after minor revision.
Author Response
Point 1: Page 1, line 25: There are no case reports of children but large cohorts for multiple centers and should be referenced.
Response 1: Thank you for you comments. “Case reports” has been corrected into “Multi-center case series”. And relevant references have been added.
Point 2: Page 3, line 66; Reference.
Response 2: Relevant references have been added.
Point 3: The authors stated that the aneurysm can be a consequence of the underlying disease as well as Covid, however difficult to tease out. There are reports of adults suffering from MIS –C syndrome however with no significant complications. This case could belong into that category.
Response 3: We totally agree with your opinion. As you mentioned, the clinical manifestations complicated by COVID-19 in adult patients might be somewhat different from those with MIS-C syndrome. Indeed, only the RCA was affected in the current study. Some comments about this issue have been added in the discussion section.
Point 4: Was the patient investigated for MIS-C? Did he receive any anti-inflammatory treatment?
Response 4: The CRP checked in our hospital was within normal range as written in the current study. And there were no findings suggesting MIS-C other than single coronary artery aneurysm. Any anti-imflammatory treatments could not be done because any confirmative evidences implying inflammation had not been detected. Some comments about this issue have been added in the discussion section.
Reviewer 2 Report
Thank you for submitting your research to Diagnostic. The topic is original and interesting, but some aspects need clarifying:
- Introduction: please report references for the statements at lines 23-26.
- Case presentation: please specify why the patient underwent a diagnostic coronary angiogram after contracting COVID-19 and how much time passed between his recovering from covid and the aneurysm finding. Moreover, I would be interested in knowing the size of the aneurysm, the indication to treatment and why a surgical bypass was necessary. Finally, I would be interested in knowing whether covid-19 infection was symptomatic or not, and if it was, what clinical manifestations it caused.
- Discussion: The causal relationship between covid-19 and the aneurysm, as you have correctly stated, could not be assessed by mere temporal coincidence. How do you think it could be proved? Which kind of further research do you propose to this purpose?
- line 80, edit "most current" to "current" or "most recent"
Author Response
Point 1: Introduction: please report references for the statements at lines 23-26.
Response 1: References have been attached.
Point 2: Please specify why the patient underwent a diagnostic coronary angiogram after contracting COVID-19 and how much time passed between his recovering from covid and the aneurysm finding.
Response 2: First, the patient had to underwent a diagnostic coronary angiography due to newly developed chest pain and a history of coronary stent implantation. To clarify why he underwent a diagnostic CAG, some words have been added. Secondly, he had been confirmed as COVID-19 four months prior to detecting the RCA aneurysm. So it had been about three months between his recovering from COVID-19 and the aneurysm finding.
Point 3: Moreover, I would be interested in knowing the size of the aneurysm, the indication to treatment and why a surgical bypass was necessary.
Response 3: The diameter of the aneurysm was 8.6 mm. In addition, it was necessary for him to undergo coronary revascularization because he had suffered from newly developed chest pain and multiple stenotic lesions were significant. He and our heart team had decided to perform surgical interventions both for the stenotic lesions and the aneurysm because percutaneous coronary interventions had not been appropriate for the coronary aneurysm. Some comments about this issue have been added.
Point 4: Finally, I would be interested in knowing whether covid-19 infection was symptomatic or not, and if it was, what clinical manifestations it caused.
Response 4: He was admitted for about 3-4 weeks due to pneumonia complicated by COVID-19. Some words have been added.
Point 5: The causal relationship between covid-19 and the aneurysm, as you have correctly stated, could not be assessed by mere temporal coincidence. How do you think it could be proved? Which kind of further research do you propose to this purpose?
Response 5: We totally agree with you opinion and it may be a limitation of this report. However, the previous CAG checked prior to suffering from COVID-19 revealed no coronary aneruysm at all and the RCA had not been touched during the CAG. COVID-19 could affect the coronary artery as well, so we need to consider the probability of COVID-19 complicating an aneursymal formation of the coronary artery. Also, we need to collect some cases concerning newly developed coronary aneurym after recent COVID-19 infection. Some comments have been added.
Point 6: Line 80, edit "most current" to "current" or "most recent"
Response 6: As your recommendation, “most current” has been corrected into “most recent”.
Reviewer 3 Report
Publish as it is
Author Response
Thank you for your decision.
Reviewer 4 Report
The manuscript can be accepted after major revision.
1. This case study is a new and unique report. The patient details have been presented well.
2. Although it is well reported that Covid 19 is primarily affect the respiratory system disorder and vascular structures. It is mentioned that the patient had been subjected to coronary stent implantation. Are the authors inferring that occurrence of the aneurysm is a result of Covid 19 infection? Or it may be due the age and previous stent implantation of the patient.
3. As the authors have correctly mentioned that cases of coronary aneurysms have been reported in adults. However, in a prominent case, COVID-19-related giant coronary aneurysms was reported in an infant (https://doi.org/10.1155/2021/8872412). Is there a reason why is it more reported in infants compared to adults?
4. A little more information may be added upon the choice of treatment (surgical intervention here) to make it easier for the non-medical readers to understand.
Overall, this one of the first reported cases to shed light upon the relationship between COVID 19 and development of coronary artery aneurysm in adults.
Author Response
Point 1: Although it is well reported that Covid 19 is primarily affect the respiratory system disorder and vascular structures. It is mentioned that the patient had been subjected to coronary stent implantation. Are the authors inferring that occurrence of the aneurysm is a result of Covid 19 infection? Or it may be due the age and previous stent implantation of the patient.
Response 1: We can only assume that COVID-19 would affect the coronary artery. However, it is clear that the right coronary artery where the aneurysm occurred had not been touched during the previous intervention because it had been clean without any stenotic lesions at that time. Some comments about this issue have been added.
Point 2: As the authors have correctly mentioned that cases of coronary aneurysms have been reported in adults. However, in a prominent case, COVID-19-related giant coronary aneurysms was reported in an infant (https://doi.org/10.1155/2021/8872412). Is there a reason why is it more reported in infants compared to adults?
Response 2: It is known that the coronary aneurysm complicated by COVID-19 in infants would results from multisystem inflammatory sydnrome (MIS-C). However, it is assumed that the mechanism of coronary aneurysm in adults would be different a bit from MIS-C considering clinical manifestations. Some comments about this issue have been added to the Discussion section.
Point 3: A little more information may be added upon the choice of treatment (surgical intervention here) to make it easier for the non-medical readers to understand.
Response 3: Thank you for your comments to improve our case. Some comments about why sugcial interventions had been selected have been added.
Round 2
Reviewer 4 Report
accepted in present form.